# Identifying the Spatio-Temporal Characteristics of Crime in Liangshan Prefecture, China

**DOI:** 10.3390/ijerph191710862

**Published:** 2022-08-31

**Authors:** Wuxue Cheng, Yajun Rao, Yixin Tang, Jiajia Yang, Yuxin Chen, Li Peng, Jiangcheng Hao

**Affiliations:** 1The Faculty Geography Resources Sciences, Sichuan Normal University, Chengdu 610101, China; 2Key Laboratory of Land Resources Evaluation and Monitoring in Southwest China, Sichuan Normal University, Chengdu 610066, China; 3Xi’an Yirun Cultural Landscape Planning and Design Co., Ltd., Xi’an 710075, China

**Keywords:** spatio-temporal evolution pattern of crime, spatio-temporal cube, property crime, violent crime, crime hotspots

## Abstract

Crime prevention and governance play critical roles in public security management. Liangshan Yi Autonomous Prefecture in Sichuan Province has a high crime rate, and spatio-temporal analysis of crime in this region could assist with public security management. Therefore, Liangshan Prefecture was selected as the research object in this study. The spatial crime data were obtained from China Judgments Online, and property crime, violent crime, and special crime (i.e., pornography, gambling, drugs, and guns) were analyzed. The findings were as follows. In terms of time characteristics (month, day, and hour), property crime tended to occur in autumn and winter, in the early month, on Wednesdays and Fridays, and at early morning. Violent crime tended to occur in winter and spring, on Mondays and Thursdays, and at night. Special crime occurred in spring and autumn, on Tuesdays, and in the daytime. In terms of spatial features, the central region of Liangshan Prefecture was the focal area for crime. There were obvious low-aggregation areas in the western region for special crime. The eastern region exhibited a high incidence of various crimes. Regarding the spatio-temporal evolution characteristics from 2013 to 2019, there were some obvious hotspots of violent and property crime in downtown and surrounding townships of Xichang City, which is the capital of Liangshan Prefecture. During the study period, the incidence of special crime has an obvious downward trend which shows that there are more new cold spots.

## 1. Introduction

Criminal acts seriously endanger the safety of individuals and society. Why study crime in poverty-stricken areas? Because these areas typically have a high incidence of crime. Such research also can promote the scientific development of public security management in poverty-stricken areas, thereby contributing to the achievement of stable economic and social development in these areas. Thus, the objective of this study is to (1) examine the spatial and temporal distribution of crimes in impoverished mountainous areas in Western China, (2) promote the comprehensive application of geographic information system technology to crime issues, and (3) test the effect of targeted poverty alleviation and the promotion of harmonious and stable development of impoverished mountainous areas. However, so far, most published research on crime geography has focused on developed areas, while less research has examined crimes in the poor western regions. At present, there are no exact standards for neighborhood time step when constructing a crime spatio-temporal cube. In order to improve the accuracy of the crime spatio-temporal cube model, this study introduced the periodogram method into the construction of the model to enable the spatio-temporal evolution analysis of crime more reasonably.

Research on spatial distribution of criminal behavior is developing rapidly, such as crime mapping which has already been widely used in the world. Fabiënne Kortas [1] examined the usage of varying scaling in hotspot mapping as an exploratory spatial data analysis technique. Timothy C. Hart [2] constructed a series of prospective hot spot maps, based on kernel density estimation. In terms of time analysis, Ratcliffe [3], Renger, [4] and other academics have conducted some simple, small-scale statistical analyses of crimes in the United States, Australia, and other developed countries. Furthermore, in spatial analysis, Brada [5] used two periods of data to explore the changing trajectory of crime hotspots, and Bill Hillier [6] used spatial syntax to explore the impact of street morphology on crime incidence.

In recent years, the usage of big data and model building for spatio-temporal analysis of criminal behavior has become a hot topic. For example, Lin Liu and colleagues drew a spatial distribution map of fraud crimes in a city and analyzed the factors that influenced fraud crimes [7]. Daqian Liu et al. studied murder, robbery, theft, and other crimes in Changchun City and used the GIS platform to draw a distribution map of crime hotspots to provide a basis for relevant departments to take effective defense measures [8]. Thangavelu A. identified the distribution of crimes to challenges facing the police departments that pursue to implement computerized crime mapping systems [9]. Giedrė Beconytė presented the results of violent crime spatial distribution in Lithuania [10]. Anthony R. Cummings used three spatial analysis tools, hotspot analysis, mean center, and standard deviation ellipse to examine the changing distribution of violent crime in Guyana [11]. Zengli Wang provide a micro view of crime distribution within hot crime areas [12]. Those above studies were all based on quantitative model analysis, which increases the practicality of research results. However, most of them did not focus on the poverty-stricken areas in Western China, and the application of mathematical models has not yet met requirements of practical use.

In summary, this study explored the spatio-temporal distribution of crime in impoverished areas in Western China. This was accomplished according to the spatial distribution and the time attributes contained in each crime location using common spatial and temporal analysis methods. This study discussed the spatio-temporal distribution of crime in poor areas in Western China to provide some support for public security management. In order to quantitatively explore the spatial-temporal evolution characteristics of crime in this region, this paper uses standardized crime intensity index, spatial autocorrelation, and crime spatio-temporal cube to summarize spatio-temporal characteristics of crime. The application of these quantitative research methods is the great significance to control crime in Liangshan Prefecture and stable development of society and economy.

## 2. Materials and Methods

### 2.1. Overview of the Study Area

Liangshan Yi Autonomous Prefecture is located in the southwest of Sichuan Province, China, between latitude 26°03′–29°18′ N and longitude 100°03′–103°52′ E (Figure 1). It starts from the Dadu River and borders Ya’an City and Ganzi Prefecture on the north; south to the Jinsha River and Yunnan Province; Zhaotong City in Yunnan Province, Yibin City and Leshan City in Sichuan Province lies on the east; and Ganzi Prefecture lies on the west. Liangshan Prefecture has 17 county-level administrative divisions under its jurisdiction and approximately 4.908 million permanent residents. Liangshan’s capital is Xichang City that is located in the central region of Prefecture. Due to outdated and backward local social concepts, poor living environments, and outdated infrastructure, Liangshan Prefecture is a typical poverty-stricken area within China, and 11 counties inhabited by ethnic minorities within its jurisdiction are all poverty-stricken counties. There are many reasons for poverty in this area, including geographical location, poor natural conditions, low education level of people, simple industrial system, etc. Among them, backward production method is the main reason for poverty. For example, there is a low rate of mechanization, and manual operation is the majority technique used. The poverty situation in these areas has further contributed to the occurrence of crime, which has become an urgent problem for local public security [13].

### 2.2. Data Source and Preprocessing

This article mainly uses the data of judgment documents published by the people’s courts of counties and cities in Liangshan Prefecture. Judgment document data comes from China Judgments Online and Wusong Com. For detailed information, see the Appendix A. In 2016, the Supreme People’s Court of the People’s Republic of China has established an Order named *Provisions of the Supreme People’s Court on People’s Courts Publishing Judgments on the Internet*. This Order clarifies that the China Judgments Online is the unified platform to publish the effective judgment documents on the internet for the all levels People’s Courts. China Judgment Online has already become the largest judicial document website in the world. Wusong Com is a company that integrate the internet industry with legal services. This study collected a total of 12,919 first-instance criminal judgments in Liangshan Prefecture from 2013 to 2019.

Data preprocessing: First, data with completely time and place information were included; 12,919 pieces of crime data were preliminarily screened out. Then, with the help of the ArcGIS 10.6 platform, 1231 pieces of data that were outside the Liangshan Prefecture were excluded. Finally, the remaining 11,688 pieces of data information were classified according to the crime types and they were divided into three categories: property crime, violent crime, and special crime (such as pornography, gambling, drugs, or guns).

### 2.3. Research Methods

#### 2.3.1. Spatial and Temporal Analysis of Geographic Data

The standardized crime intensity index was used to analyze the occurrence intensity of different crimes within a certain period of time. Thus, the temporal differentiation characteristics of the crimes were obtained. The degree of clustering of different crimes in the area was analyzed by spatial autocorrelations to determine the spatial characteristics of the area.

#### 2.3.2. Crime Spatio-Temporal Cube Model

The working principle of the spatio-temporal cube model can be roughly summarized as follows. First, the time intercept point features are used to construct a net CDF data cube by aggregating the points into space-time bins. The cube structure has rows, columns, and time steps. If you multiply the number of rows by the number of columns and time steps, you will get the total number of bins in the cube. The rows and columns determine spatial extent of the cube, and the time steps determine temporal extent. Referring to the methods of Andong Hong, Zechun Huang, Xuemin Hu, and others, the distance interval and time step of crime spatio-temporal cube was obtained by respectively using spatial analysis theory and the periodic research.

In the choice of distance interval, the spatial autocorrelation method was used to study the relationship between spatial clustering degree and distance range. The spatial autocorrelation of geographic event points was measured at a range of distances, and a scatterplot of these distances and the corresponding z-scores was drawn. The corresponding fitted curve was used to obtain the z-score at any interval distance, and the z-score reflects the degree of spatial clustering. How interesting is that statistically significant peaks may exist in multiple peaks, and peaks associated with larger distances usually reflect a broad trend.

In the choice of time step, periodogram method was used to calculate the implicit period of crime. For a certain type of event that changes with time, this method combines experimental period and actual sequence to explore the implicit period. The calculation process includes three steps: original sequence linear trend test, implicit period calculation, and significance test. The specific steps are as follows:

(1) The original sequence linear trend test: the unit root test is performed on the original sequence. If the original sequence is not stationary, its linear trend is removed, and a new sequence without a linear trend is obtained and tested again until the sequence appears stationary.

(2) Implicit period calculation: a function of the form is decomposed into a Fourier series, where the Fourier coefficients of the first harmonic are:(1)$$A_{T}=2/K_{T}\sum_{t=1}^{K_{T}}Z_{t}\cos2\pi t/T$$
(2)$$B_{T}=2/K_{T}\sum_{t=1}^{K_{T}}Z_{t}\sin2\pi t/T$$

In the formula, *K* is the maximum integer multiple of *T* included in the sequence length, *T* is the period, *Z_t_* is the number of cases per day, *A_T_* and *B_T_* are the Fourier coefficients of *Z_t_*, and *T* is the period station, $S_{T}^{2}=A_{T}^{2}+B_{T}^{2}$; when $S_{T}^{2}$ is the maximum value, the *T* value may be the implicit period of the sequence.

(3) Significance test: in order to exclude false implicit period caused by random fluctuations, the maximum value needs to be tested for significance. The formula is as follows:(3)$$J=S_{T}^{2}KT/4\sigma^{2}$$
σ^2^ is the variance of the original time sequence. When *J* ≥ Ja (*J*0.01 = 4.605), the confidence level is 99% and it is considered that there is an implicit period in the crime time sequence.

#### 2.3.3. Evolution of Spatio-Temporal Patterns of Hot and Cold Spots

The Emerging Spatio-temporal Hotspot Analysis tool can identify trends in the data. For example, it can discover new, intensified, reduced, and scattered hot and cold spots. The Space-time Net CDF cube was used as the input, which was created by the Create Spatio-temporal cube tool from aggregate points. Then, neighborhood distances and neighborhood time steps long parameter value was used to calculate Getis-Ord Gi* statistic (hot spot analysis) for each bin. After completing the Emerging Spatio-temporal Hotspot Analysis, each bin in the Net CDF cube has an associated z-score, p-value, and added hotspot bin classification. Then, we evaluated these hot and cold spot trends by using the Mann–Kendall trend test, which was proposed by Khaled H. Hamed [14]. Based on each location with data, trend z-scores and *p*-values and hotspots for each bin were generated. If a feature has a high z-score and a small *p*-value, there is a spatial aggregation of high values (hot spot); if the z-score is negative number with a low value and the *p*-value is small, there is a spatial aggregation of low values (cold spot); if the z-score is close to 0, there is no obvious spatial aggregating.

## 3. Results

### 3.1. Time Distribution Characteristics of Crime

By examining the pre-processed data and using the standardized crime index to quantify crime intensity [15,16], we can obtain an intuitive view of property crime, violent crime, and special crime over the time scale (month, day, and hour) (Figure 2, Figure 3, Figure 4 and Figure 5). The specific findings are described below.

For property crime:(1)Monthly distribution: January to March and August to December were identified as high incidence periods for property crime; April to July was a low incidence period of property crime.(2)Daily distribution: Friday had the largest number of incidents in a week, while Sunday had the least. Wednesday and Friday were two peaks of crime incidents during the week. In general, the average number on weekdays was greater than weekends. The 1st day has the largest number, and the 31st day has the least. There was a clear contrast between the first two days and the last two days in a month. Except the first two days, the number of property crimes are within a low range in early month. The number of crimes in middle month are within a high range, and in late month basically within a low range.(3)Hour distribution: The number of property crimes at 2 A.M. was the highest, and 7 A.M. was the lowest. There were three peak points in the distribution within a day: 2 A.M., 10 A.M., and 9 P.M. The highest incidence period of property crime was 6 P.M. to 3 A.M. From 3 A.M. to 7 A.M., the numbers gradually decreased to a minimum, and during daytime the numbers were below the average level.

For violent crime:(1)Monthly distribution: November had the largest number, and August had the least. January to April and November to December were high incidence periods of violent crime, and May to October was a low incidence period.(2)Daily distribution: Same as property crime, Friday had the largest number of violent crime and Sunday had the least. Wednesday and Friday were still two peaks during the week. Similarly, the average number on weekdays was greater than weekends. Here there is a difference: the number of violent crimes on the 22nd was the largest, and on the 31st was the least.(3)Hourly distribution: The highest number of violent crimes occurred at 11 P.M., and the least number at 5 A.M. There were two peak points: 11 P.M. and 1 A.M. Those two points were significantly higher than at other times, which is an abnormally high value area at these times.

For special crime:(1)Monthly distribution: March had a higher number than any other months, and it was an abnormally high value month. June, July, and December had the least. January to May and September to December were high incidence periods of special crime. Among them, April was an abnormal low number within a high incidence period. Meanwhile, June to August was a low incidence period.(2)Daily distribution: Unlike property and violent crime, Tuesday had the largest number of special crime, while Sunday had the least. It is roughly decreased from Monday to Sunday. The number of incidents on the 1st was the largest, and the number of incidents on the 31st was the least. Except those two days, the number of cases basically remained stable. Overall, the number of special crime fluctuated less than the other crime types, and it was close to the average number.(3)Hourly distribution: The largest number of special crime occurred at 3 P.M., and the least number at 5 A.M. Four peak points, namely 10 A.M., midday, 3 P.M., and 9 P.M., are shown in the distribution. This generally indicates that more crimes occur during daytime and fewer at nighttime.

### 3.2. Spatial Distribution Characteristics of Crime

GeoDa software was used to run a global autocorrelation analysis on the spatial distribution of crimes, roads, and public facilities in Liangshan Prefecture [17,18,19]. The results are shown in Table 1. From 2013 to 2019, the spatial distribution of various crime types exhibited certain aggregation characteristic, no obvious discrete trend, and a positive spatial correlation pattern. The size of the Z value reflects the research objects’ strength of the spatial aggregation. Among the three types of crime, property crime’s degree of agglomeration was slightly higher, and violent crime and special crime was followed.

In order to further study the spatial distribution of violent crime, property crime, and special crime in Liangshan Prefecture, GeoDa software was used to run univariate local autocorrelation analyses (Figure 6). The spatial distribution of crimes in Liangshan Prefecture was as follows: Towns within the central region were typical high-frequency crime gathering areas. Obviously, in the western region, only special crime shows low frequency gathering areas. The eastern region was more complicated. Specifically, the high frequency gathering areas of property crime, violent crime, and special crime was alternately distributed here.

### 3.3. Pattern Evolution

The parameter of neighborhood time step in Emerging Spatio-temporal Hotspot Analysis is related to the final Spatio-temporal evolution pattern expression. The autocorrelation function is used to represent the degree of correlation between observations, and the formula is as follows:(4)$$I\left(K\right)=\frac{n}{\sum_{i=1}^{n}\left(X_{i}-\bar{X}\right)}\times \frac{\sum_{i=K+1}^{n}\sum_{j=i-K}^{i-1}\left(X_{i}-\bar{X}\right)\left(X_{j}-\bar{X}\right)}{\left(n-K\right)K}$$

In the formula, *K* = 1, 2, 3... *n* − 2. When *K* got larger, *I*(*K*) would be larger, the difference between the observations in the sequence would be smaller, and the degree of aggregation would be higher. When *I*(*K*) = 0, it means that the number of event’s occurrences was unrelated to spatio-temporal evolution.

In order to explore the spatio-temporal evolution pattern of crimes from 2013 to 2019, Emerging Spatio-temporal Hotspot Analysis was performed. First, it was necessary to select an appropriate distance interval and time step to construct the crime spatio-temporal cube. The global autocorrelation analysis method was used to obtain the spatial aggregation degree (Z value) of the crime sites at different distances, as shown in Figure 7. There were two peaks in the Z value, 3.6073 and 3.6143, respectively, corresponding to distances of 1700 m and 4500 m. The largest peak was evident at 4500 m, so 4500 m was selected as the distance interval parameter of Liangshan Prefecture crime spatio-temporal cube.

In choosing the time step, periodogram method was used to calculate the implicit period of crime. Months, weeks, and days were used as the basic time intervals. Because the time sequence for months and weeks did not pass the significance test and exhibited no obvious periodicity, days was adopted as the time interval to form a time sequence of incident numbers. As shown in Figure 8, when *T* = 32, $S_{T}^{2}$max = 9.8820, *K* = 79, and got *J* > 4.605. It is considered that the original sequence had a periodic vibration, in which the first implicit period is 32 days. So, 32 days was taken as the time step parameter for Liangshan Prefecture crime spatio-temporal cube.

The number of crimes was grouped according to the time step of 32 days in the spatio-temporal cube above. In the time sequence data ADF test results, t statistic value was −1.459 and *p* value was 0.554. As *p* = 0.554 > 0.1, the null hypothesis cannot be rejected and the sequence animations are not smooth. According to the autocorrelation function formula, let us discuss the relationship between Z scores and lag period. It can be seen from Figure 9 that when the lag period was 18 steps, i.e., 576 h, Z score was the highest at 3.5518. The difference between sequence observations was the smallest, and the degree of aggregation was the most. Thus, a neighborhood time step of 18 steps was chosen. The corresponding neighborhood distances were automatically generated by the ArcGIS 10.6 platform based on the relevant parameters of the spatio-temporal cube. The neighborhood distances of violent crime, property crime, and special crime were 27,744 m, 16,415 m, and 23,348 m, respectively.

As shown in Figure 10a, there were 53 locations with cold or hot spot trends in the spatio-temporal evolution pattern of property crime. All the spatio-temporal evolution pattern of this crime were hotspots, among which, continuous hotspots took 51 of 53; the continuous hotspots were mainly distributed in Xichang City and its surrounding areas. The other two hotspots were scattered hotspots, which were located in Xichang City. Generally speaking, the number of property crime remained at a high level in Xichang City. This not only is related to its relatively developed economy, convenient shopping, and strong population mobility, but also its lack of strong external supervision. Hospitals, bus stops, pedestrian streets, and other areas where people gathering are common locations for property crimes [20].

In the spatio-temporal evolution pattern of violent crime (Figure 10b), there were 57 locations with cold or hot spot trends [21,22,23]. The crime spatio-temporal evolution model was dominated by hotspots, including two modes: continuous hotspots and oscillating hotspots. There were 23 continuous hotspots, mainly distributed in the central of region. A total of 26 oscillating hotspots which were mainly distributed the same as continuous hotspots, with the most in Xichang City. Among the cold spots, there were only seven scattered cold spots, which were distributed in the southwest region. In general, the number of violent crimes in regional counties and cities exhibited a more obvious trend over time than in the eastern and western regions [24,25].

Regarding the spatio-temporal evolution pattern of special crime (Figure 10c), there were 265 locations with cold or hot spot trends. The spatio-temporal evolution mode of crime was dominated by cold spots, which mainly included three modes: oscillating cold spots, scattered cold spots, and newly added cold spots. Through all three modes, the focus should be oscillating cold spots and new cold spots. Among them, there were 150 oscillating cold spots and only 1 new cold spot distributed in the eastern of region., and 99 scattered cold spots, mainly in the south and southwest. In summary, it is the complete opposite to violent crime and property crime; there was no significant spatio-temporal pattern in special crimes in Xichang City. However, for some counties south and east to Xichang City, there were obvious cold spot trends, especially oscillating cold spots frequently occurred. In these areas, the incidence of special crime has an obvious downward trend.

During the research period, the spatio-temporal evolution patterns in violent crime and property crime exhibited some obvious hotspots; Xichang City and surrounding counties had a contrast situation. There were obvious hotspots in the downtown area and surrounding townships. It was significate that the spatio-temporal evolution patterns of special crime shows downtrend in Liangshan Prefecture. In recent years, vigorous supervisions and targeted poverty alleviation have been implemented in this region, and it is one of the Liangshan Prefecture’s enormous successes for people living and working in peace and contentment.

## 4. Discussion

There is multitudinous literature on criminal activities in the economically prosperous in Eastern China, while less research has examined crimes in poor Western China. For example, Minling Zeng et al. studied criminal activity in Pudong New Area, Shanghai, China [26]. Shanhe Jiang et al. explored perceptions of property crime at the neighborhood level in in Guangzhou, China [27]. Liangshan Prefecture is China’s largest area inhabited by the Yi ethnic group and one of the deeply impoverished “Three Districts and Three Prefectures” areas. Until the eve of the fight against poverty, Liangshan was still one of the most deeply impoverished areas in China. A total of 11 of 17 counties in the prefecture are key counties for poverty alleviation. As a high incidence and deep degree of poverty, the study of criminal activities in Liangshan Prefecture provides a representative view of crime in impoverished areas in China.

This study examined the spatio-temporal characteristics of property crime, violent crime, and special crime in Liangshan Prefecture, and analyzed the cold and hot spots in the spatio-temporal evolution patterns of various crimes [28,29,30]. In addition, using preprocessed data, attribute analysis was performed on criminal suspects based on several factors such as gender, age, crime location, and education level. The characteristics of criminal suspects that engaged in property crime, violent crime, and special crime in Liangshan Prefecture between 2013 and 2019 were as follows.

The gender and age compositions of suspects involved in violent and property crime were similar; these suspects were mainly men in their 30s with a generally low level of education. Violent crime suspects tended to be younger. The suspects involved in special crime had a high crime rate, were older in age, and were more often female. The sources of suspects in the central and southern of Liangshan Prefecture were more complex. Criminal suspects in the eastern of region tended to go out to commit crimes, which is related to the economic development of each county. The criminal suspects in all cases generally had low education level; less than 10% of the suspects had a high school education or above. Chuhong Wang found that college expansion causally reduces crime rates and effect changes over time [31], which confirms that education can play a significant positive role in crime prevention.

Therefore, in our next research phase, we will consider the impact of education on criminal activities [32,33,34]. In addition, due to the availability and limitations of the current data, this study was unable to conduct a longer time sequence analysis of the dynamics of criminal activities. In the future, we plan to predict the spatial and temporal distribution of criminal activities according to the characteristics of crimes in the western region, and establish a special model [35,36,37] in order to provide more scientific and effective management tools for public security management in Western China. A safer environment includes the usage of modern governance techniques and means, such as facial recognition technology, and normalized inspection. Local governments will formulate crime prevention and governance policies in accordance with the requirements of central government and the actual local situation. It also includes some special actions to reduce crime, forming good social benefits.

## 5. Conclusions

Based on this research on crime in Liangshan Prefecture from 2013 to 2019, the following conclusions can be drawn:(1)Property crime, violent crime, and special crime in Liangshan Prefecture exhibited obvious temporal differences. The various types of crime exhibited different patterns over the month, day, and hour scales. Property crime mainly occurred in autumn and winter; in early month; on Wednesdays and Fridays; and at early morning. Violent crime mainly occurred in winter and spring; on Mondays and Thursdays; and at night. Special crime mainly occurred in the spring and autumn; on Tuesdays; and in the daytime.(2)The property crime, violent crime, and special crime in Liangshan Prefecture exhibited obvious spatial differences. Towns within the central region area were typical high-frequency crime gathering areas, with higher numbers of violent crime, property crime, and special crime. Obviously, in the western region, only special crime shows low frequency gathering areas. The situation in the eastern region was more complicated. Specifically, the high frequency gathering areas of property crime, violent crime, and special crime was alternately distributed here.(3)Cases of property crime, violent crime, and special crime in Liangshan Prefecture exhibited obvious spatio-temporal pattern evolution characteristics. From 2013 to 2019, the spatio-temporal evolution patterns of violent crime and property crime exhibited obvious hotspot distributions; Xichang City and surrounding counties had a contrast situation. There were obvious hotspots in the downtown area and its surrounding areas. Due to national policies and public security management, it was significate that the spatio-temporal evolution patterns of special crime shows downtrend in Liangshan Prefecture.

## Figures and Tables

**Figure 1 ijerph-19-10862-f001:**
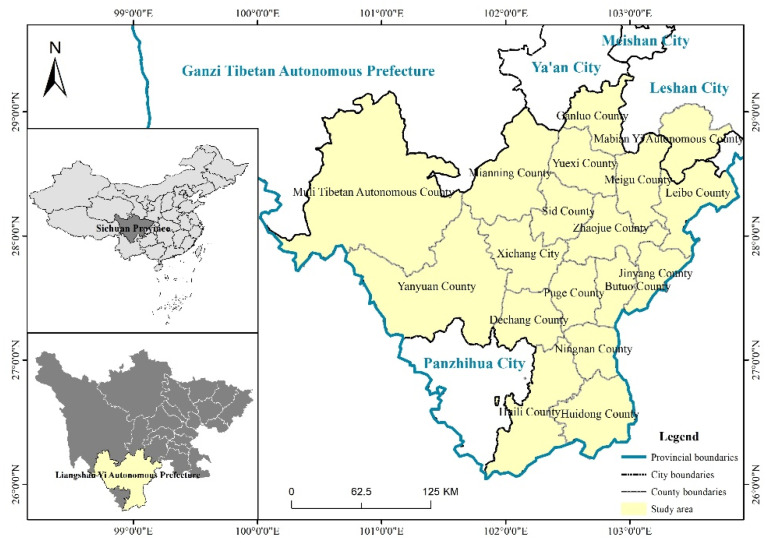
Location of the study area.

**Figure 2 ijerph-19-10862-f002:**
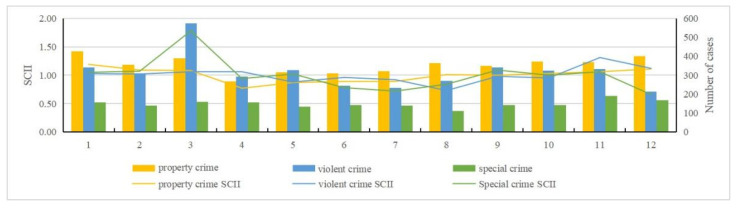
Standardized monthly crime intensity index and number of cases.

**Figure 3 ijerph-19-10862-f003:**
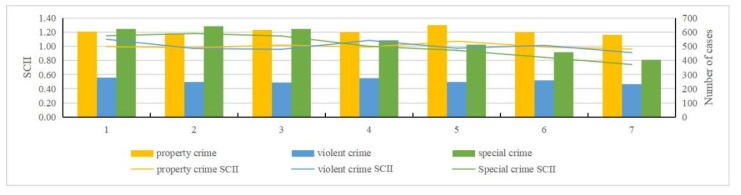
Standardized weekly crime intensity index and number of cases.

**Figure 4 ijerph-19-10862-f004:**
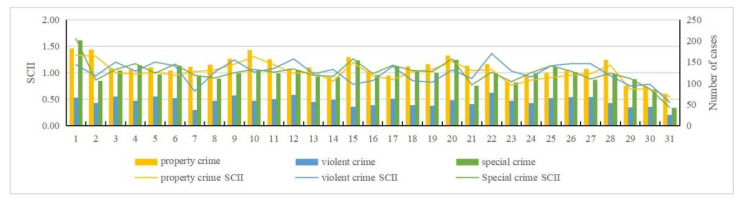
Standardized day scale crime intensity index and number of cases.

**Figure 5 ijerph-19-10862-f005:**
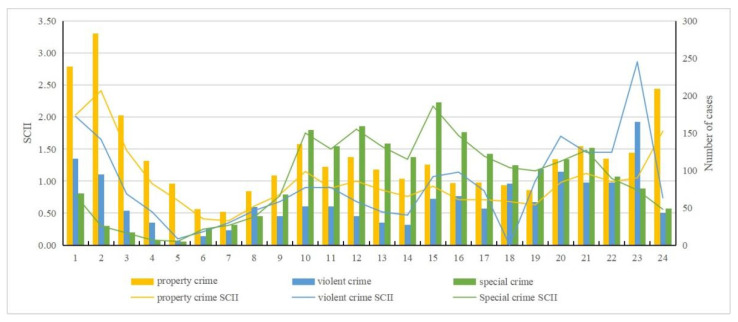
Standardized hourly scale crime intensity index and number of cases.

**Figure 6 ijerph-19-10862-f006:**
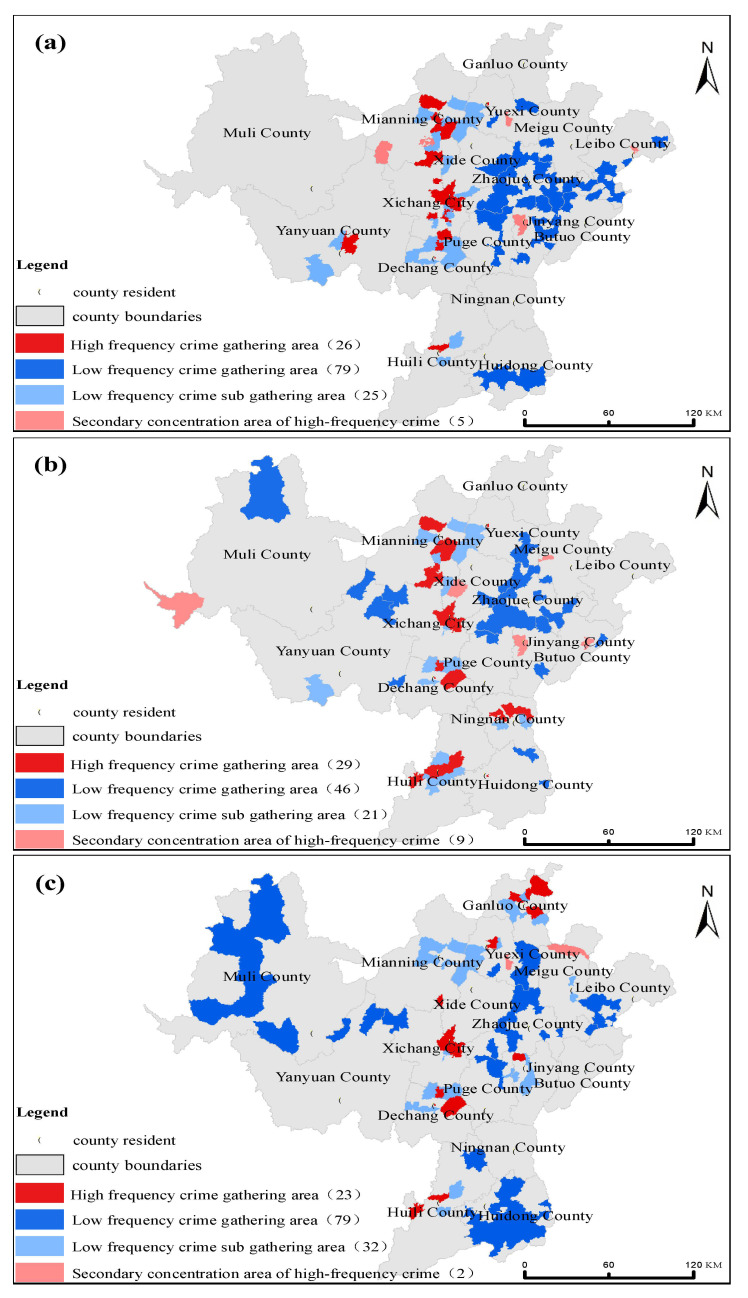
Local spatial autocorrelation results for the various crime: (**a**) Property crime; (**b**) Violent crime; (**c**) Special crime; The figures in brackets represent the number of crimes of all kinds.

**Figure 7 ijerph-19-10862-f007:**
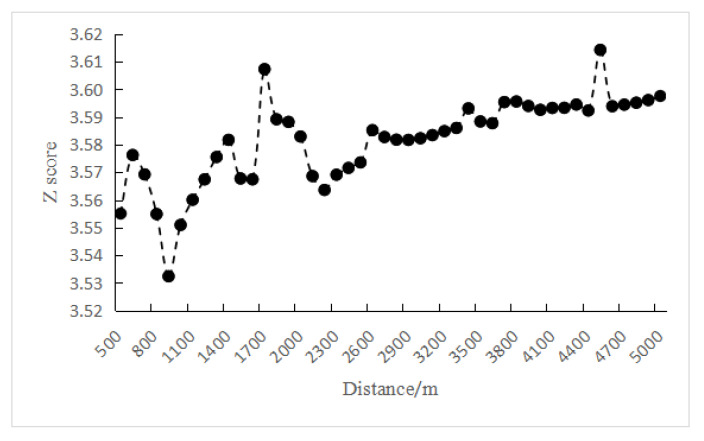
Global spatial autocorrelation Z scores of crime point distribution.

**Figure 8 ijerph-19-10862-f008:**
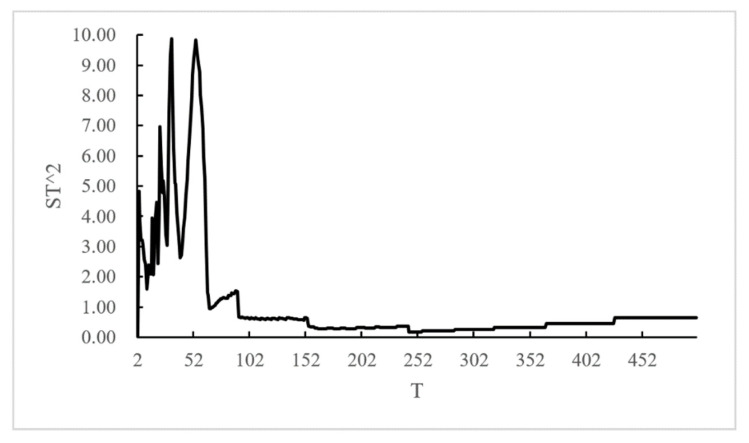
Incident time cycle.

**Figure 9 ijerph-19-10862-f009:**
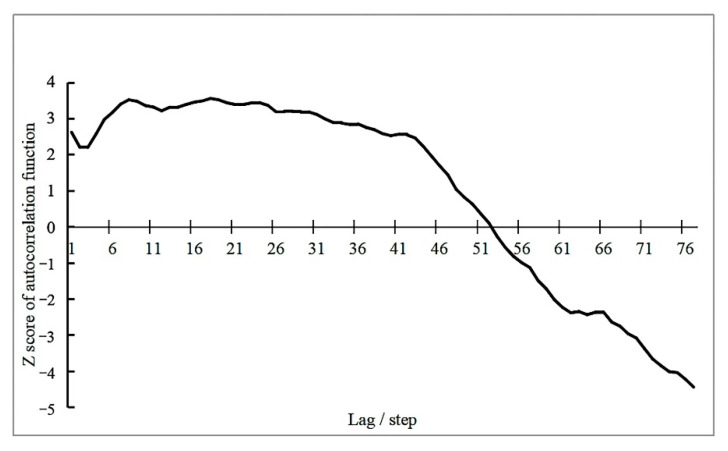
Time sequence sample autocorrelation function Z score diagram.

**Figure 10 ijerph-19-10862-f010:**
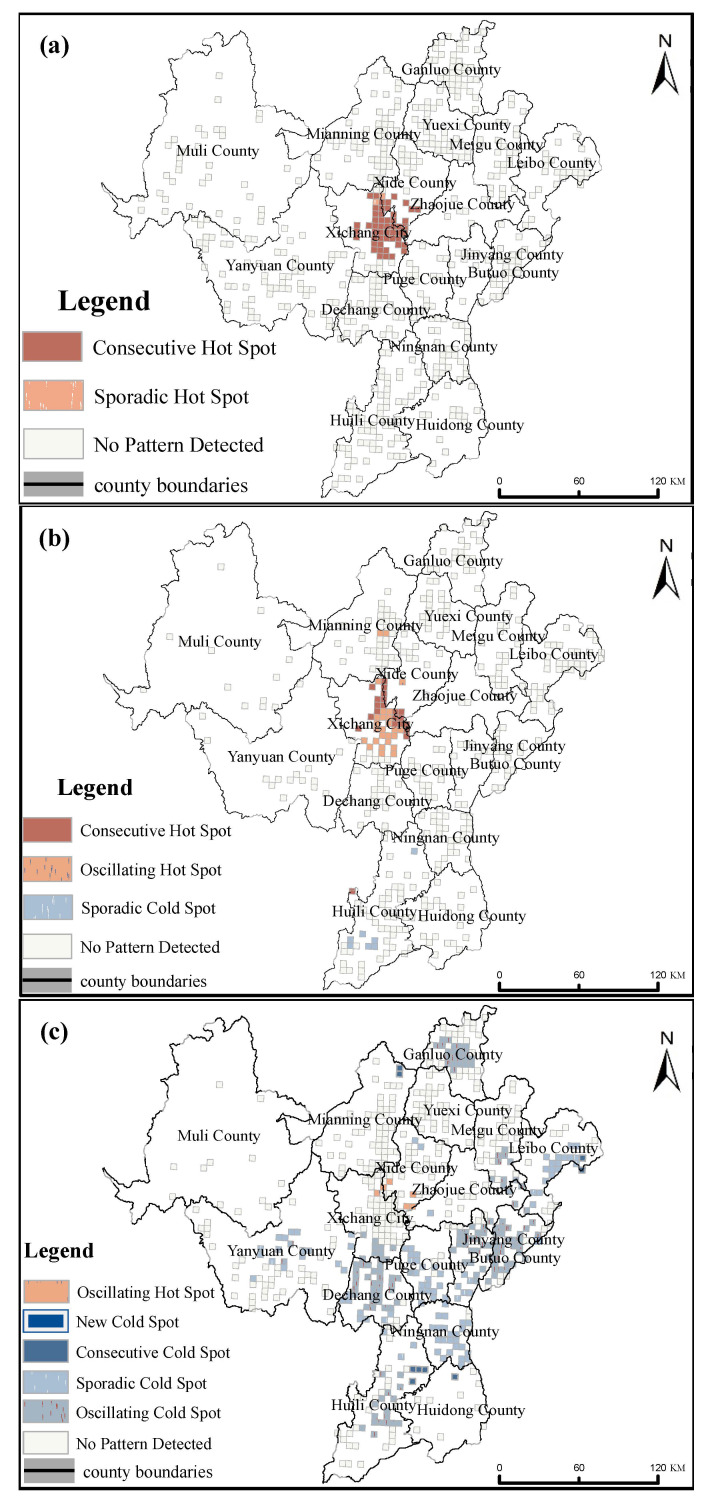
Spatio-temporal evolution pattern of crime: (**a**) Property crime; (**b**) Violent crime; (**c**) Special crime.

**Table 1 ijerph-19-10862-t001:** Spatial autocorrelation results of various crime.

Type of Crime	Z	I	P
property crime	6.1161	0.0678	0.0060
Violent crime	5.3507	0.0877	0.0040
Special crime	4.8380	0.0845	0.0060

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
