# Peer review of "Identifying the Spatio-Temporal Characteristics of Crime in Liangshan Prefecture, China"

_ijerph, 2022, doi:10.3390/ijerph191710862_

Round 1

Reviewer 1 Report

The paper has potential. But further detailed work is required.

Reviewer 2 Report

The article deals with the important issue of the spatial distribution of crime, but it does so in a cursory, in places careless way. It contains significant deficiencies or errors that require supplementing or correction.

Firstly, it is not legitimate to generalize one city's case study as "mountain crimes". And this is the goal that the Authors of the article set for themselves, without referring to it later either in the discussion or in the conclusions.

Secondly, the introduction makes no reference to the key literature on the spatial distribution of crime. It is not true that the development of this type of crime research method took place only in the 21st century - research has been conducted since the 19th century, which is confirmed by numerous pieces of literature. In the 21st century, GIS-based tools are used for this purpose, which the authors omit from the theoretical introduction.

Thirdly, the analyzed data relating to the period before COVID-19, but has nowhere been taken as a research assumption. So it can be assumed that the authors were simply able to obtain such data. Meanwhile, it is known from numerous studies conducted in this area that the pandemic has had and has an impact on the spatial distribution of crime. In the light of the above, the proposed research is of a historical nature.

Fourthly, the discussion section practically does not refer to the literature on the subject. Poor bibliography in the context of the number of publications available on the topic in question.

Fifth, very poor quality figure 6 - maps and descriptions are practically unreadable.

I recommend major revision and extensive editing of English. In my opinion, without introducing the recommended changes, the article should be rejected

Round 2

Reviewer 1 Report

The paper is much better now. The extra references and the additional explanations strengthen the paper.  The link with theory is also much better.

Reviewer 2 Report

After the introduced corrections, the article presents a more complete picture of the analyzed problem. However, the theoretical part still does not contain reference to the key items in the field of crime mapping.
